# Dynamics of Supervised and Reinforcement Learning in the Non-Linear Perceptron

**Christian Schmid**
Institute of Neuroscience
University of Oregon
`cschmid9@uoregon.edu`

**James M. Murray**
Institute of Neuroscience
University of Oregon
`jmurray9@uoregon.edu`

## Abstract

The ability of a brain or a neural network to efficiently learn depends crucially on both the task structure and the learning rule. Previous works have analyzed the dynamical equations describing learning in the relatively simplified context of the perceptron under assumptions of a student-teacher framework or a linearized output. While these assumptions have facilitated theoretical understanding, they have precluded a detailed understanding of the roles of the nonlinearity and input-data distribution in determining the learning dynamics, limiting the applicability of the theories to real biological or artificial neural networks. Here, we use a stochastic-process approach to derive flow equations describing learning, applying this framework to the case of a nonlinear perceptron performing binary classification. We characterize the effects of the learning rule (supervised or reinforcement learning, SL/RL) and input-data distribution on the perceptron's learning curve and the forgetting curve as subsequent tasks are learned. In particular, we find that the input-data noise differently affects the learning speed under SL vs. RL, as well as determines how quickly learning of a task is overwritten by subsequent learning. Additionally, we verify our approach with real data using the MNIST dataset. This approach points a way toward analyzing learning dynamics for more-complex circuit architectures.

## Introduction

Learning, which is typically implemented in both biological and artificial neural networks with iterative update rules that are noisy due to the noisiness of input data and possibly of the update rule itself, is characterized by stochastic dynamics. Understanding these dynamics and how they are affected by task structure, learning rule, and neural-circuit architecture is an important goal for designing efficient artificial neural networks (ANNs), as well as for gaining insight into the means by which the brain's neural circuits implement learning.

As a step toward developing a full mathematical characterization of the dynamics of learning for multilayer ANNs solving complex tasks, recent work has made progress by making simplifying assumptions about the task structure and/or neural-circuit architecture. One fruitful approach has been to study learning dynamics in what is perhaps the simplest non-trivial ANN architecture: the individual perceptron. Even with this simplification, however, fully characterizing the mathematics of learning has been challenging for complex tasks, and further simplifications have been required.

38th Conference on Neural Information Processing Systems (NeurIPS 2024).

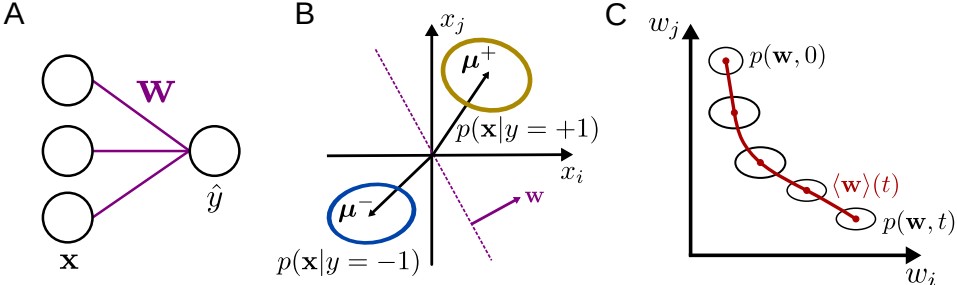

Figure 1: Learning dynamics in the nonlinear perceptron. **A:** The perceptron, parametrized by weights $\mathbf{w}$, maps an input $\mathbf{x}$ to the output $\hat{y}$. **B:** The inputs are drawn from two multivariate normal distributions with labels $y = \pm 1$. The weight vector $\mathbf{w}$ is orthogonal to the classification boundary. **C:** Due to the stochasticity inherent in the update equations, the weights are described by the flow of a probability distribution in weight space.

One approach has been to analyze learning in the student-teacher framework [Gardner and Derrida, 1989, Seung et al., 1992], in which a student perceptron learns to produce an input-to-output mapping that approximates that of a teacher perceptron. This has led to insights about the differences in learning dynamics between different types of learning rules (*e.g.*, supervised and reinforcement learning) [Werfel et al., 2003, Züge et al., 2023, Patel et al., 2023]. Such insights are highly relevant for neuroscience, where a longstanding goal has been to infer the learning mechanisms that are used in the brain [Lim et al., 2015, Nayebi et al., 2020, Portes et al., 2022, Humphreys et al., 2022, Mehta et al., 2023, Payeur et al., 2023]. However, by construction, the student-teacher setup in the perceptron only applies to input-output mappings that are linearly separable, which is seldom the case in practice. Another approach has been to study learning dynamics in the linearized perceptron [Werfel et al., 2003, Mignacco et al., 2020, Bordelon and Pehlevan, 2022], which enables exact solutions even for structured input data distributions that are not linearly separable. However, the dynamics of learning in nonlinear neural networks—even very simple ones—performing classification tasks are not fully understood. Further, whether and how the dynamics of learning might differ under different learning rules in such settings has not been investigated.

Here, we take a stochastic-process approach (similar to Yaida [2018] and Murray and Escola [2020]) to derive flow equations describing learning in the finite-dimensional nonlinear perceptron trained in a binary classification task (Fig. 1). These results are compared for two different online learning rules: supervised learning (SL, which corresponds to logistic regression) and reinforcement learning (RL). We characterize the effects of the input-data distribution on the learning curve, finding that, for SL but not for RL, noise along the coding direction slows down learning, while noise orthogonal to the coding direction speeds up learning. In addition, we verify our approach by training a nonlinear perceptron on the MNIST dataset. Finally, applying the approach to continual learning, we quantify how the input noise and learning rule affect the rate at which old classifications are forgotten as new ones are learned. Together, these results establish the validity of the approach in a simplified context and provide a path toward analyzing learning dynamics for more-complex tasks and architectures.

## Stochastic-process approach for describing weight evolution

We consider a general iterative update rule of the form

$$w_i^{t+\delta t} - w_i^t = \eta f_i(\mathbf{w}^t), \tag{1}$$

where $\mathbf{w}^t \in \mathbb{R}^n$ for arbitrary $n > 0$, and $\eta$ is the learning rate. The stochastic update term $f_i$ on the right-hand side is drawn from a probability distribution—it depends on the weights themselves, as well as the input to the network and, potentially, output noise. Starting from this update equation, our goal is to derive an expression characterizing the evolution of the probability distribution of the weights, $p(\mathbf{w}, t)$ (cf. Fig. 1C). We assume that $f_i(\mathbf{w})$ does not explicitly depend on $\eta$, and that all the moments $\langle f_i^k \rangle_L$, $k = 1, 2, \ldots$, where $\langle \cdot \rangle_L$ denotes an average over the noise in the update equation (1) (including the input distribution as well as, potentially, output noise), exist as smooth functions of $\mathbf{w}$.

Given the stochastic process defined by (1), the probability distribution at time $t + \delta t$ given the distribution at time $t$ is

$$p(\mathbf{w}, t + \delta t) = \int d\mathbf{w}' p(\mathbf{w}, t + \delta t | \mathbf{w}', t) p(\mathbf{w}', t). \tag{2}$$

Denoting the weight update as $\delta \mathbf{w} := \mathbf{w} - \mathbf{w}'$, the integrand in this equation can be written as

$$p(\mathbf{w}, t + \delta t | \mathbf{w}', t) p(\mathbf{w}', t) = p(\mathbf{w} + \delta \mathbf{w} - \delta \mathbf{w}, t + \delta t | \mathbf{w} - \delta \mathbf{w}, t) p(\mathbf{w} - \delta \mathbf{w}, t). \tag{3}$$

Changing the integration variable to $\delta \mathbf{w}$ and performing a Taylor expansion in $\delta \mathbf{w}$, the right-hand side of (2) yields

$$\int d\mathbf{w}' p(\mathbf{w}, t + \delta t | \mathbf{w}', t) p(\mathbf{w}', t) =$$
$$p(\mathbf{w}, t) - \sum_i \frac{\partial}{\partial w_i} [\alpha_i(\mathbf{w}) p(\mathbf{w}, t)] + \frac{1}{2} \sum_{ij} \frac{\partial^2}{\partial w_i \partial w_j} [\beta_{ij}(\mathbf{w}) p(\mathbf{w}, t)] + O(\delta \mathbf{w}^3), \tag{4}$$

where

$$\alpha_i(\mathbf{w}) = \int d\delta \mathbf{w} \, \delta w_i \, p(\mathbf{w} + \delta \mathbf{w}, t + \delta t | \mathbf{w}, t) \tag{5}$$

and

$$\beta_{ij}(\mathbf{w}) = \int d\delta \mathbf{w} \, \delta w_i \delta w_j \, p(\mathbf{w} + \delta \mathbf{w}, t + \delta t | \mathbf{w}, t). \tag{6}$$

Here, we assumed that the probability distribution describing the weight updates $f$ has bounded derivatives with respect to $\mathbf{w}$.

Although (2) is only defined for discrete time steps, we assume a continuous probability density $p(\mathbf{w}, t)$ interpolates between the updates and exists as a smooth function for all values of $t$. We can then expand the left-hand side of (2) to obtain

$$p(\mathbf{w}, t + \delta t) = p(\mathbf{w}, t) + \delta t \frac{\partial}{\partial t} p(\mathbf{w}, t) + O(\delta t^2). \tag{7}$$

For the iterative update rules that we will consider, we have $\delta \mathbf{w} \propto \eta$, where $\eta$ is a learning rate. In order to take a continuous-time limit, we let $\eta := \delta t$ and take the limit $\delta t \to 0$. For the general learning rule (1), the coefficients in (4) have the form

$$\alpha_i(\mathbf{w}) = \langle f_i \rangle_L, \quad \beta_{ij}(\mathbf{w}) = \langle f_i f_j \rangle_L, \tag{8}$$

where $\langle \cdot \rangle_L$ denotes an average over the noise in the update equation (1) (including the input distribution as well as, potentially, output noise). Thus, we find

$$\eta \frac{\partial p}{\partial t}(\mathbf{w}, t) = -\eta \sum_i \frac{\partial}{\partial w_i} (p(\mathbf{w}, t) \langle f_i \rangle_L) + \mathcal{O}(\eta^2). \tag{9}$$

Finding the $p(\mathbf{w}, t)$ that solves this equation cannot in general be done exactly when $f_i$ is nonlinear. However, by multiplying (9) with powers of $\mathbf{w}$ and integrating, as well as expanding in $\mathbf{w} - \langle \mathbf{w} \rangle$, where $\langle \cdot \rangle$ denotes the average with respect to $p(\mathbf{w}, t)$, we can derive a system of equations for the moments of $p(\mathbf{w}, t)$ [Risken, 1996]. As we derive in the appendix, this gives the following expressions for the first two moments up to $\mathcal{O}((\mathbf{w} - \langle \mathbf{w} \rangle)^3)$:

$$\frac{d}{dt} \langle w_i \rangle = \left( 1 + \frac{1}{2} \sum_{k,l} \mathrm{Cov}(w_k, w_l) \partial_k \partial_l \right) \langle f_i \rangle_L(\langle \mathbf{w} \rangle), \tag{10}$$

$$\frac{d}{dt} \mathrm{Cov}(w_i, w_j) = \sum_k [\mathrm{Cov}(w_i, w_k) \partial_k \langle f_j \rangle_L(\langle \mathbf{w} \rangle) + \mathrm{Cov}(w_j, w_k) \partial_k \langle f_i \rangle_L(\langle \mathbf{w} \rangle)]. \tag{11}$$

Together, these equations characterize the flow of $p(\mathbf{w}, t)$ for a general iterative learning algorithm in a general ANN architecture.

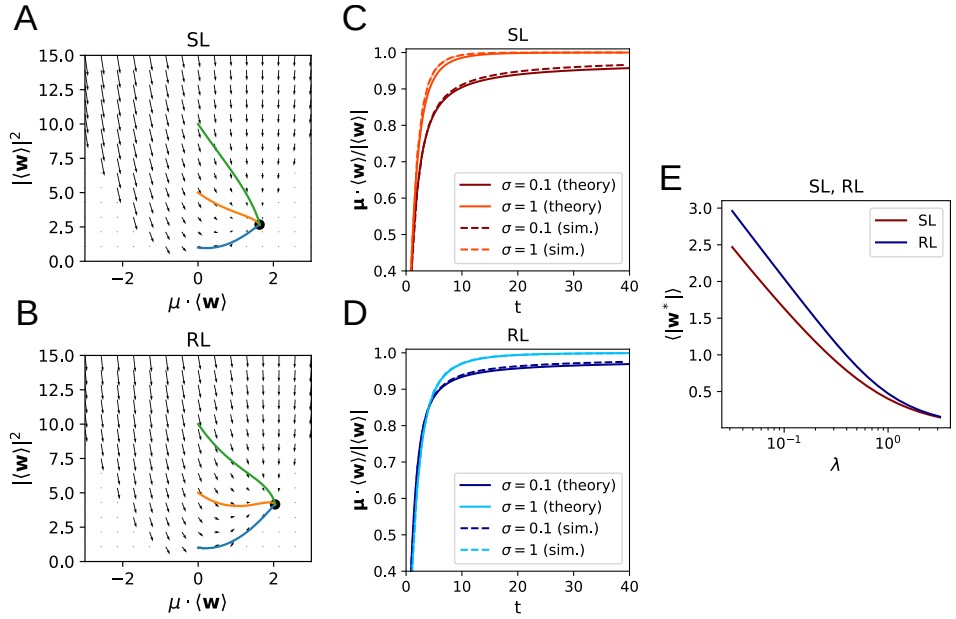

Figure 2: Learning dynamics in a perceptron classification task. **A, B**: Flow fields determining the weight dynamics with trajectories for different initial conditions for SL (A) and RL (B). **C, D**: Learning dynamics from simulations closely follow the analytical results for SL (C) and RL (D). **E**: Dependence of the asymptotic weight norm on the regularization parameter $\lambda$.

## Learning dynamics in the nonlinear perceptron

While the above approach is general and could be applied to any iterative learning algorithm for any ANN architecture, for the remainder of this work we will focus on its application to the nonlinear perceptron (Fig. 1A), a one-layer neural network that receives an input $\mathbf{x} \in \mathbb{R}^N$, multiplies it with a weight vector $\mathbf{w} \in \mathbb{R}^N$, and produces an output $\hat{y}$. The task we study is a binary Gaussian classification task, in which the model is presented with samples $\mathbf{x}$ drawn from two distributions $p(\mathbf{x}|y)$ with labels $y = \pm 1$, where $p(y = \pm 1) = \frac{1}{2}$. Each $p(\mathbf{x}|y)$ is given by a multivariate normal distribution with $\mathbf{x} \sim \mathcal{N}(\boldsymbol{\mu}^y, \boldsymbol{\Sigma}^y)$ (Fig. 1B). We analyze both the case of SL with deterministic output, for which $\hat{y} = \phi(\mathbf{w} \cdot \mathbf{x})$, as well as RL, for which the stochastic output is given by $\pi(\hat{y} = \pm 1) = \phi(\pm \mathbf{w} \cdot \mathbf{x})$, where $\phi$ is the logistic sigmoid function. The goal of the model is to output a label $\hat{y}$ that closely matches the ground truth $y$ when given an input $\mathbf{x}$.

### Derivation of the flow equations

The supervised learning rule we consider is regularized stochastic gradient descent for a binary cross-entropy loss, which results in the weight update rule

$$f(\mathbf{w})_i = (\tilde{y} - \hat{y})x_i - \lambda w_i, \tag{12}$$

where $\tilde{y} = \frac{1}{2}(y + 1) \in \{0, 1\}$ is the shifted input label, and $\lambda$ is the regularization hyperparameter. This learning rule describes online logistic regression.

For reinforcement learning, we use the REINFORCE policy-gradient rule with reward baseline [Williams, 1992, Sutton and Barto, 2018]:

$$f(\mathbf{w})_i = \hat{y}\delta\phi(-\hat{y}\mathbf{w} \cdot \mathbf{x})x_i - \lambda w_i. \tag{13}$$

Here $\delta = y\hat{y} - \langle y\hat{y} \rangle$ is the reward prediction error, and $\hat{y}$ is the stochastic output of the perceptron with probability $\pi(\hat{y} = \pm 1) = \phi(\pm \mathbf{w} \cdot \mathbf{x})$. To facilitate mathematical feasibility, we replace the perceptron activation function by a shifted error function $\phi(z) = \frac{1}{2}\left(1 + \text{Erf}\left(\frac{\sqrt{\pi}}{4}z\right)\right)$.

We first derive the learning dynamics for stochastic gradient descent. We assume that the initial condition is uniquely specified, with $p(\mathbf{w}, 0) = \delta(\mathbf{w} - \mathbf{w}^0)$. In this case, the weight covariance will

be zero, and the flow equations (10) simply reduce to

$$\frac{d}{dt}\langle w_i\rangle = \langle f_i\rangle_L\big|_{\mathbf{w}=\langle\mathbf{w}\rangle}. \tag{14}$$

To make the formulas more concise, we set $\lambda = 0$. It can be reintroduced by simply adding the term $-\lambda\mathbf{w}$. We then get

$$\langle f_i\rangle_L(\mathbf{w}) = \langle(\tilde{y}-\phi(\mathbf{w}\cdot\mathbf{x}))x_i\rangle_{\mathbf{x},y}$$

$$= \frac{1}{2}\langle(1-\phi(\mathbf{w}\cdot\mathbf{x}))x_i\rangle_{\mathbf{x}\sim\mathcal{N}(\boldsymbol{\mu}^+,\boldsymbol{\Sigma}^+)} - \frac{1}{2}\langle\phi(\mathbf{w}\cdot\mathbf{x})x_i\rangle_{\mathbf{x}\sim\mathcal{N}(\boldsymbol{\mu}^-,\boldsymbol{\Sigma}^-)}$$

$$= \frac{1}{2}\mu_i^+\left(1-\Phi\left(\frac{a_+}{\sqrt{1+b_+^2}}\right)\right) - \frac{1}{2}\frac{1}{\sqrt{2\pi}}\frac{(\boldsymbol{\Sigma}^+\cdot\tilde{\mathbf{w}})_i}{\sqrt{1+b_+^2}}e^{-\frac{a_+^2}{2(1+b_+^2)}} \tag{15}$$

$$-\frac{1}{2}\mu_i^-\Phi\left(\frac{a_-}{\sqrt{1+b_-^2}}\right) - \frac{1}{2}\frac{1}{\sqrt{2\pi}}\frac{(\boldsymbol{\Sigma}^-\cdot\tilde{\mathbf{w}})_i}{\sqrt{1+b_-^2}}e^{-\frac{a_-^2}{2(1+b_-^2)}}.$$

Here, $\Phi(z) = \frac{1}{2}\left(1+\mathrm{Erf}\left(z/\sqrt{2}\right)\right) = \phi(z\cdot\sqrt{8/\pi})$ is the cumulative distribution function of the standard normal distribution. To simplify notation, we have introduced $\tilde{\mathbf{w}} = \mathbf{w}\cdot\sqrt{\pi/8}$, as well as the quantities

$$a_y = \boldsymbol{\mu}^y\cdot\tilde{\mathbf{w}}, \tag{16}$$

$$b_y = \sqrt{\tilde{\mathbf{w}}^T\boldsymbol{\Sigma}^y\tilde{\mathbf{w}}}. \tag{17}$$

To aid interpretation of these results, we assume that $\boldsymbol{\mu}^\pm = \pm\boldsymbol{\mu}$ and $\boldsymbol{\Sigma} = \sigma^2\mathbb{I}$. Then (15) implies

$$\frac{d}{dt}\langle\boldsymbol{\mu}\cdot\mathbf{w}\rangle = |\boldsymbol{\mu}|^2\left(1-\Phi\left(\frac{\boldsymbol{\mu}\cdot\tilde{\mathbf{w}}}{\sqrt{1+\sigma^2|\tilde{\mathbf{w}}|^2}}\right)\right) - \frac{1}{\sqrt{2\pi}}\frac{\sigma^2\boldsymbol{\mu}\cdot\tilde{\mathbf{w}}}{\sqrt{1+\sigma^2|\tilde{\mathbf{w}}|^2}}e^{-\frac{(\boldsymbol{\mu}\cdot\tilde{\mathbf{w}})^2}{2(1+\sigma^2|\tilde{\mathbf{w}}|^2)}}\Bigg|_{\mathbf{w}=\langle\mathbf{w}\rangle} \tag{18}$$

as well as

$$\frac{d}{dt}|\langle\mathbf{w}\rangle|^2 = 2\mathbf{w}\cdot\boldsymbol{\mu}\left(1-\Phi\left(\frac{\boldsymbol{\mu}\cdot\tilde{\mathbf{w}}}{\sqrt{1+\sigma^2|\tilde{\mathbf{w}}|^2}}\right)\right) - \frac{1}{2}\frac{\sigma^2|\mathbf{w}|^2}{\sqrt{1+\sigma^2|\tilde{\mathbf{w}}|^2}}e^{-\frac{(\boldsymbol{\mu}\cdot\tilde{\mathbf{w}})^2}{2(1+\sigma^2|\tilde{\mathbf{w}}|^2)}}\Bigg|_{\mathbf{w}=\langle\mathbf{w}\rangle} \tag{19}$$

An interpretation of (18) is that the first term pushes the weight vector in the decoding direction, while the second term acts as a regularization, whereby the cross-entropy loss penalizes misclassifications more as $\boldsymbol{\mu}\cdot\mathbf{w}$ increases. An increase in the input noise leads to a higher overlap of the distributions, which means that even the Bayes-optimal classifier will make more mistakes.

For RL, we need to calculate

$$\langle f_i\rangle_L(\mathbf{w}) = \langle\hat{y}\delta\phi(-\hat{y}\mathbf{w}\cdot\mathbf{x})x_i\rangle_{\mathbf{x},y,\hat{y}}$$

$$= \langle\phi(-\mathbf{w}\cdot\mathbf{x})\phi(\mathbf{w}\cdot\mathbf{x})x_i\rangle_{\mathbf{x}\sim\mathcal{N}(\boldsymbol{\mu}^+,\boldsymbol{\Sigma}^+)} - \langle\phi(-\mathbf{w}\cdot\mathbf{x})\phi(\mathbf{w}\cdot\mathbf{x})x_i\rangle_{\mathbf{x}\sim\mathcal{N}(\boldsymbol{\mu}^-,\boldsymbol{\Sigma}^-)}$$

$$= \frac{(\boldsymbol{\Sigma}^+\cdot\tilde{\mathbf{w}})_i}{\sqrt{2\pi}\sqrt{1+b_+^2}}e^{-\frac{a_+^2}{2(1+b_+^2)}}\left(1-2\Phi\left(\frac{a_+}{\sqrt{1+b_+^2}\sqrt{1+2b_+^2}}\right)\right) + 2\mu_i^+ T\left(\frac{a_+}{\sqrt{1+b_+^2}},\frac{1}{\sqrt{1+2b_+^2}}\right) \tag{20}$$

$$- \frac{(\boldsymbol{\Sigma}^-\cdot\tilde{\mathbf{w}})_i}{\sqrt{2\pi}\sqrt{1+b_-^2}}e^{-\frac{a_-^2}{2(1+b_-^2)}}\left(1-2\Phi\left(\frac{a_-}{\sqrt{1+b_-^2}\sqrt{1+2b_-^2}}\right)\right) - 2\mu_i^- T\left(\frac{a_-}{\sqrt{1+b_-^2}},\frac{1}{\sqrt{1+2b_-^2}}\right).$$

Here, $T(\cdot,\cdot)$ is Owen's T function:

$$T(h,a) = \frac{1}{2\pi}\int_0^a \frac{e^{-\frac{1}{2}h^2(1+x^2)}}{1+x^2}dx. \tag{21}$$

As for supervised learning, we can simplify this expression for isotropic distributions with means $\pm\boldsymbol{\mu}$ and get

$$\frac{d}{dt}\langle\boldsymbol{\mu}\cdot\mathbf{w}\rangle = |\boldsymbol{\mu}|^2 4T\left(\frac{\boldsymbol{\mu}\cdot\tilde{\mathbf{w}}}{\sqrt{1+\sigma^2|\tilde{\mathbf{w}}|^2}},\frac{1}{\sqrt{1+2\sigma^2|\tilde{\mathbf{w}}|^2}}\right)$$

$$- \frac{1}{\sqrt{2\pi}}\frac{2\sigma^2\boldsymbol{\mu}\cdot\tilde{\mathbf{w}}}{\sqrt{1+\sigma^2|\tilde{\mathbf{w}}|^2}}e^{-\frac{(\boldsymbol{\mu}\cdot\tilde{\mathbf{w}})^2}{2(1+\sigma^2|\tilde{\mathbf{w}}|^2)}}\,\mathrm{Erf}\left(\frac{\boldsymbol{\mu}\cdot\tilde{\mathbf{w}}}{\sqrt{1+\sigma^2|\tilde{\mathbf{w}}|^2}\sqrt{2+4\sigma^2|\tilde{\mathbf{w}}|^2}}\right)\Bigg|_{\mathbf{w}=\langle\mathbf{w}\rangle} \tag{22}$$

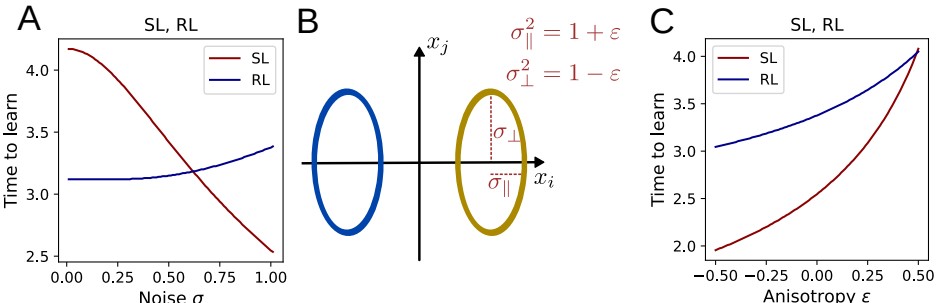

Figure 3: Relationship between input noise and time to learn the task. **A:** The time required for the alignment $\boldsymbol{\mu} \cdot \langle \mathbf{w} \rangle / |\langle \mathbf{w} \rangle|$ to reach 80% depends on the noise $\sigma$ of the isotropic input distributions. **B:** To characterize anisotropic input noise, the total input variance is split into a noise component $\sigma_\parallel^2$ parallel to and a component $\sigma_\perp^2$ orthogonal to the decoding direction. **C:** Shifting the input noise into the decoding direction slows down learning.

and

$$
\begin{aligned}
\frac{d}{dt} |\langle \mathbf{w} \rangle|^2 = {} & 8\mathbf{w} \cdot \boldsymbol{\mu} T \left( \frac{\boldsymbol{\mu} \cdot \tilde{\mathbf{w}}}{\sqrt{1 + \sigma^2 |\tilde{\mathbf{w}}|^2}}, \frac{1}{\sqrt{1 + 2\sigma^2 |\tilde{\mathbf{w}}|^2}} \right) \\
& - \frac{1}{2} \frac{2\sigma^2 |\mathbf{w}|^2}{\sqrt{1 + \sigma^2 |\tilde{\mathbf{w}}|^2}} e^{-\frac{(\boldsymbol{\mu} \cdot \tilde{\mathbf{w}})^2}{2(1 + \sigma^2 |\tilde{\mathbf{w}}|^2)}} \operatorname{Erf} \left( \frac{\boldsymbol{\mu} \cdot \tilde{\mathbf{w}}}{\sqrt{1 + \sigma^2 |\tilde{\mathbf{w}}|^2} \sqrt{2 + 4\sigma^2 |\tilde{\mathbf{w}}|^2}} \right) \Big|_{\mathbf{w} = \langle \mathbf{w} \rangle}.
\end{aligned}
\tag{23}
$$

As we show in the appendix, and as demonstrated in Fig. 2, the flow equations for both SL and RL have a unique, globally stable fixed point whenever $\lambda > 0$ or the input noise $\sigma > 0$ (Fig. 2A,B). The solutions of (15) and (20) exhibit agreement with learning curves obtained by direct simulation of (1) (Fig. 2C,D), where the small remaining discrepancy arises from the fact that, for the simulation, we used a standard logistic sigmoid function instead of the error function sigmoid curve used for the analytical calculations. We also see that the asymptotic weight norm decreases approximately linearly with $\ln \lambda$ (Fig. 2E). Of particular note is the observation that, perhaps counter-intuitively, higher levels of noise appear to lead to *faster* learning for SL, though the effect is more ambiguous in the case of RL. This will be analyzed in more detail in the following section.

**Impact of noise on learning time**

We next investigate the effect of different types of input noise on the dynamics of learning and whether differences arise for the supervised and reinforcement algorithms. We begin with the case of isotropic input noise, with $\boldsymbol{\Sigma} = \sigma^2 \mathbb{I}$ and means $\pm \boldsymbol{\mu}$ with $|\boldsymbol{\mu}| = 1$. In this case, the optimal alignment $\frac{\boldsymbol{\mu} \cdot \langle \mathbf{w} \rangle}{|\langle \mathbf{w} \rangle|}$ of 1 is always reached asymptotically, so we focus on how quickly this value is approached as a function of the input noise.

In the case of SL, analytically analyzing the logarithmic derivative of the alignment between $\boldsymbol{\mu}$ and $\langle \mathbf{w} \rangle$ yields a flow equation of the form

$$
\frac{d}{dt} \log \frac{\boldsymbol{\mu} \cdot \langle \mathbf{w} \rangle}{|\langle \mathbf{w} \rangle|} = g_{\mathrm{iso}}(\boldsymbol{\mu}, \mathbf{w}) + \sigma^2 h_{\mathrm{iso}}(\boldsymbol{\mu}, \mathbf{w})^2 + \mathcal{O}(\sigma^4),
\tag{24}
$$

where $g_{\mathrm{iso}}$ and $h_{\mathrm{iso}}$ do not depend on $\sigma$. Thus, the higher the input noise, the faster the task is learned. The analogous relationship for RL is indeterminate, such that input noise may either speed up or slow down learning in this case, depending on the parameters. As is illustrated in Fig. 3A, numerical integration of the flow equations reveals qualitatively distinct trends for the dependence of learning speed on noise.

**Anisotropic input distributions**

To analyze the case of anisotropic input noise, we divide the total noise into two components: a component $\sigma_\parallel^2 = 1 + \varepsilon$ in the direction of $\boldsymbol{\mu}$ and the noise $\sigma_\perp^2 = 1 - \varepsilon$ orthogonal to it, while keeping

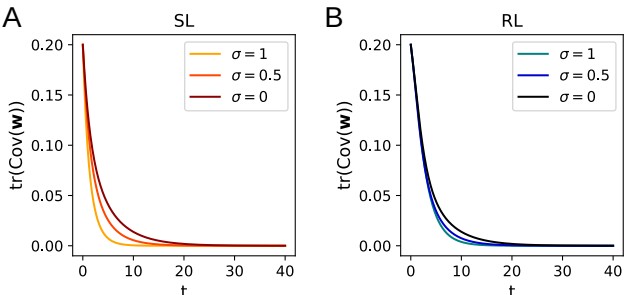

Figure 4: Dynamics of the total variance of $\mathbf{w}$ for isotropic input noise. Higher noise leads to a faster decay in $\mathrm{tr}\left(\mathrm{Cov}(\mathbf{w})\right)$ for supervised learning (**A**) and for reinforcement learning (**B**).

the total noise $\sigma_\parallel^2 + \sigma_\perp^2$ fixed (Fig. 3B). For both SL and RL, we find that learning slows down when the noise is shifted to the decoding direction and speeds up when it is shifted to orthogonal directions (Fig. 3C). To confirm this analysis analytically, we calculate the logarithmic derivative of the alignment between $\boldsymbol{\mu}$ and $\langle \mathbf{w} \rangle$ and find

$$\frac{d}{dt} \log \frac{\boldsymbol{\mu} \cdot \langle \mathbf{w} \rangle}{|\langle \mathbf{w} \rangle|} = g_{\mathrm{an}}(\boldsymbol{\mu}, \mathbf{w}) + \varepsilon h_{\mathrm{an}}(\boldsymbol{\mu}, \mathbf{w})^2 + \mathcal{O}(\varepsilon^2), \tag{25}$$

where $g_{\mathrm{an}}$ and $h_{\mathrm{an}}$ are independent of $\varepsilon$. From this expression, we see that, at least to leading order in $\varepsilon$, noise anisotropy orthogonal to the decoding direction tends to increase the speed of learning, while anisotropy along the decoding direction tends to decrease the speed of learning. This is in apparent contrast to a recent study in two-layer networks, where input variance along the task-relevant dimension was found to *increase* the speed of learning [Saxe et al., 2019]. The reason for these seemingly opposite results is because, in the the task studied in that work, variance along the coding direction is a signal that facilitates learning, while, in our case of binary classification, variance along the coding direction is noise that impairs learning.

**Input noise covariance**

So far, we have assumed that the initial weight distribution, which can be thought of as characterizing an ensemble of networks with different initializations, is specified deterministically, *i.e.* $p(\mathbf{w}, 0) = \delta(\mathbf{w} - \mathbf{w}^0)$. In this case, according to (11), the covariance of $\mathbf{w}$ will remain zero at later times. If training is instead initiated with a distribution $p(\mathbf{w}, 0)$ having nonzero covariance, then we can ask how this covariance evolves with training—in particular, whether the covariance of this distribution diverges, converges to 0, or approaches a finite value as $t \to \infty$.

This calculation can be easily performed in the limit $\sigma \to 0$ where the inputs are just $x = \pm\boldsymbol{\mu}$. Then (15) simply becomes

$$\langle f_i \rangle_L(\mathbf{w}) = \mu_i \left(1 - \phi(\boldsymbol{\mu} \cdot \mathbf{w})\right) - \lambda w_i, \tag{26}$$

and (11) implies that

$$\frac{d}{dt} \mathrm{tr}\left(\mathrm{Cov}(\mathbf{w})\right) = -\frac{e^{-\pi(\boldsymbol{\mu}\cdot\mathbf{w})^2/16}}{4} \boldsymbol{\mu}^T \mathrm{Cov}(\mathbf{w})\boldsymbol{\mu} - 2\lambda \mathrm{tr}\left(\mathrm{Cov}(\mathbf{w})\right). \tag{27}$$

Since $\mathrm{Cov}(\mathbf{w})$ is positive semidefinite, both terms on the right-hand side of (27) are always nonpositive for $\lambda > 0$ and lead to exponential decay of $\mathrm{tr}\left(\mathrm{Cov}(\mathbf{w})\right)$, so the eigenvalues of $\mathrm{Cov}(\mathbf{w})$ approach zero. Thus, the covariance of the distribution $p(\mathbf{w}, t)$ vanishes as $t \to \infty$ (Fig. 4A).

The same calculation can be performed for the RL algorithm, again with the result that $\mathrm{tr}(\mathrm{Cov}(\mathbf{w})) \to 0$ as $t \to \infty$ whenever $\lambda > 0$ (Fig. 4B). As can be seen in Fig. 4, the total variance continues to decay to zero upon including input noise (in the $\eta \to 0$ limit we are working in), with the decay speeding up as the noise is increased.

**Application to real tasks**

In order to test whether the theoretical equations derived above apply to realistic input data, we next train a perceptron with stochastic gradient descent to perform binary classification with cross-

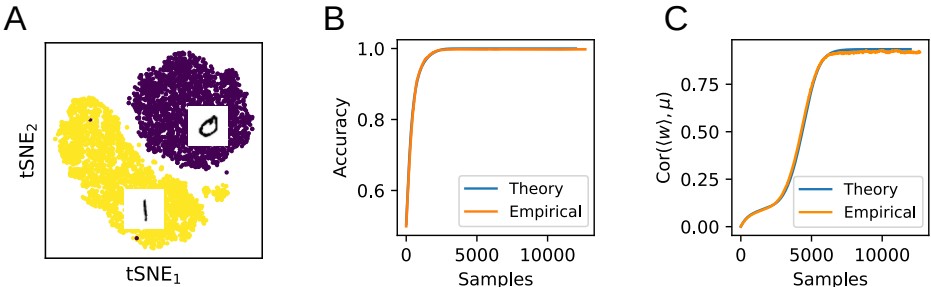

Figure 5: Comparison of the theory with training on MNIST. **A:** A nonlinear perceptron is trained to classify the digits 0 and 1 in the MNIST dataset. **B:** Comparison of the empirical test classification accuracy with the theoretical prediction. **C:** Even after the task has been learned, the theory accurately captures non-trivial ongoing learning dynamics.

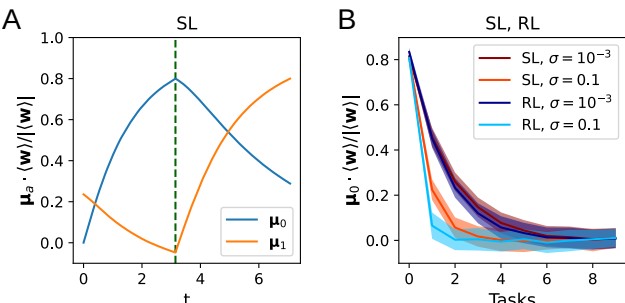

Figure 6: Forgetting curves. **A**: Learning curves for multi-task learning, where $\mathbf{w}$ are trained on Task 1 ($\boldsymbol{\mu} = \boldsymbol{\mu}_1$) after training to 80% on Task 0 ($\boldsymbol{\mu} = \boldsymbol{\mu}_0$). **B**: The alignment of $\langle \mathbf{w} \rangle$ with $\boldsymbol{\mu}_0$ after training on additional tasks $1, \ldots, 9$.

entropy loss on the MNIST dataset (Fig. 5A). To obtain suitable input representations, the images corresponding to the digits 0 and 1 are first convolved with a set of 1440 Gabor filters [Haghighat et al., 2015]. (In the appendix, we perform the same analysis on the raw MNIST data without the Gabor convolution and obtain similar results.) We then model these two input classes as multivariate Gaussians with covariances $\boldsymbol{\Sigma}_{0,1}$ and means $\boldsymbol{\mu}_{0,1}$ (or $\pm\boldsymbol{\mu}$ after a translation). The evolution of the weight vector during training is found by numerically integrating (15). To quantify the test accuracy during training, an approximation of the expected error at each time step is derived by integrating the Gaussian approximations to the two input distributions up to the hyperplane orthogonal to the weight vector. As can be seen in Fig. 5B, this theoretically derived learning curve closely matches the actual generalization performance of the trained classifier on the hold-out set.

To further illustrate that the flow equations capture non-trivial aspects of the learning dynamics, Fig. 5C shows the alignment of $\mathbf{w}$ with $\boldsymbol{\mu}$, which continues to evolve after the task has been learned. The close alignment of the experimental results with the analytical predictions shows that the flow equations can capture learning dynamics in a realistic task with input data distributions that are not necessarily Gaussian.

**Continual learning**

In addition to describing the dynamics of learning a single task, the flow equations derived above can also describe the learning and forgetting of multiple tasks. In continual learning, natural and artificial agents struggle with catastrophic forgetting, which causes older learning to be lost as it is overwritten with newer learning [Hadsell et al., 2020, Kudithipudi et al., 2022, Flesch et al., 2023]. Here, we ask how the number of tasks that can be remembered by the perceptron depends on the level of noise and the learning algorithm. The weights are first trained on Task 0, with input distribution defined by $\boldsymbol{\mu} = \boldsymbol{\mu}_0$ and $\boldsymbol{\Sigma} = \sigma^2 \mathbb{I}$, until the alignment of $\mathbf{w}$ with $\boldsymbol{\mu}_0$ has reached 80%. We then train on subsequent tasks $\boldsymbol{\mu} = \boldsymbol{\mu}_1, \boldsymbol{\mu}_2, \ldots$. This yields a forgetting curve that decays exponentially

with the number of tasks, as shown in the simulation results in Fig. 6. The decay constant does not significantly depend on the learning algorithm being used, but we observe that a higher input noise leads to faster forgetting. Together with the results in the preceding subsections, this hints toward a trade-off between the learning speed and forgetting of previously learned tasks as the amount of input noise is varied.

## Discussion

In this work, we have used a stochastic-process framework to derive the dynamical equations describing learning in the nonlinear perceptron performing binary classification. We have quantified how the input noise and learning rule affect the speed of learning and forgetting, in particular finding that greater input noise leads to faster learning for SL but not for RL. Finally, we have verified that our approach captures learning dynamics in an MNIST task that has a more-complex input data distribution. Together, the results characterize ways in which task structure, learning rule, and neural-circuit architecture significantly impact learning dynamics and forgetting rates.

One limitation of our approach is the assumption that the input distributions are multivariate Gaussians, which may not be the case for real datasets. While the agreement between the theoretical and empirical results applied to the MNIST data in Fig. 5 is encouraging in this regard, there may be greater discrepancies in cases where the input distributions are more complex. Indeed, recent work on the nonlinear perceptron has shown that, while the first- and second-order cumulants of the input distribution are learned early in training, later stages of training involve learning beyond-second-order (i.e. non-Gaussian) statistical structure in the input data [Refinetti et al., 2023], suggesting that our theory's ability to describe late-stage training in complex datasets may be somewhat limited. Another limitation is the choice to neglect higher-order terms in $\mathbf{w} - \langle \mathbf{w} \rangle$ (Equations (10), (11)) and $\eta$ (Equation (9)). This may limit the ability to characterize instabilities and noise effects induced by non-infinitesimal learning rates. Future work will be needed to assess these effects.

While other work has approached SGD learning in neural networks within a stochastic-process framework, most of these works have not derived the noise statistics from the noisy update rule (as done here and in Yaida [2018] and Murray and Escola [2020]), but rather have added Gaussian noise to the mean update (*e.g.* [He et al., 2019, Li et al., 2019, 2021]). While the results for the flow of the weights' mean $\langle \mathbf{w} \rangle(t)$ are the same under both approaches, the approach that we take enables us to additionally derive the flow of the weight covariance. Further, it allows for the possibility of describing effects arising from finite learning rate by including higher-order terms in $\eta$ from the expansion of (4)—a topic that we will address in an upcoming publication.

In our results on continual learning, we found that only a few tasks could be remembered by the perceptron before being overwritten. This is perhaps somewhat surprising given recent work [Murray and Escola, 2020] showing that the binary perceptron can recall $O(N)$ individual random patterns in a continual-learning setup. This difference may arise in part from the fact that that work used a more efficient, margin-based supervised learning rule [Crammer et al., 2006] rather than the stochastic gradient descent rule used here, as well as the fact that input noise and weight regularization were not included. This difference suggests that there is likely room for significant improvements in continual-learning performance with the setup studied here. This would be another interesting direction for future work, given that recent work has found that nonlinearity can drastically increase the amount of catastrophic forgetting in continual learning [Dominé et al., 2023].

Finally, we speculate that qualitative differences between learning rules such as that shown in Fig. 3 may provide a path for designing experiments to distinguish between learning rules implemented in the brain. More work will be needed, however, to formulate testable experimental predictions for more-realistic learning rules and network architectures. More generally, the approach developed here paves the way for analyzing numerous questions about learning dynamics in more-complex circuit architectures and diverse task structures.

## Acknowledgements

We are grateful to Elliott Abe for early collaboration related to this project. Support for this work was provided by NIH-BRAIN award RF1-NS131993.

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

## Derivation of the general evolution equations

In this section, we derive equations (10) and (11) from (9). We start with

$$\frac{\partial p}{\partial t}(\mathbf{w}, t) = -\sum_j \frac{\partial}{\partial w_j} \left( p(\mathbf{w}, t) \langle f_j \rangle_L \right), \tag{28}$$

multiply both sides by $w_i$ and integrate over $\mathbf{w}$. The left-hand side simply becomes

$$\frac{d}{dt} \langle w_i \rangle_{\mathbf{w}}. \tag{29}$$

For the right-hand side, we can use integration by parts to get

$$-\int d\mathbf{w}\, w_i \sum_j \frac{\partial}{\partial w_j} \left( p(\mathbf{w}, t) \langle f_j \rangle_L \right) = \int d\mathbf{w}\, p(\mathbf{w}, t) \langle f_i \rangle_L = \langle \langle f_i \rangle_L(\mathbf{w}) \rangle_{\mathbf{w}}. \tag{30}$$

To evaluate this expectation value, we introduce the mean-zero weight $\hat{\mathbf{w}} = \mathbf{w} - \langle \mathbf{w} \rangle$, which describes the fluctuations of $\mathbf{w}$ around its mean. If we expand $\langle f_i \rangle_L(\mathbf{w})$ to second order in $\hat{\mathbf{w}}$, (30) becomes

$$\langle \langle f_i \rangle_L(\mathbf{w}) \rangle_{\mathbf{w}} = \left\langle \langle f_i \rangle_L(\langle \mathbf{w} \rangle) + \sum_j \hat{w}_j \partial_j \langle f_i \rangle_L(\langle \mathbf{w} \rangle) + \frac{1}{2} \sum_{j,k} \hat{w}_j \hat{w}_k \partial_j \partial_k \langle f_i \rangle_L(\langle \mathbf{w} \rangle) + \mathcal{O}(\hat{\mathbf{w}}^3) \right\rangle_{\mathbf{w}}$$

$$= \langle f_i \rangle_L(\langle \mathbf{w} \rangle) + \sum_j \langle \hat{w}_j \rangle_{\mathbf{w}} \partial_j \langle f_i \rangle_L(\langle \mathbf{w} \rangle) + \frac{1}{2} \sum_{j,k} \langle \hat{w}_j \hat{w}_k \rangle_{\mathbf{w}} \partial_j \partial_k \langle f_i \rangle_L(\langle \mathbf{w} \rangle) + \mathcal{O}(\hat{\mathbf{w}}^3) \tag{31}$$

$$= \langle f_i \rangle_L(\langle \mathbf{w} \rangle) + \frac{1}{2} \sum_{j,k} \mathrm{Cov}(w_k, w_j) \partial_j \partial_k \langle f_i \rangle_L(\langle \mathbf{w} \rangle) + \mathcal{O}(\hat{\mathbf{w}}^3).$$

The derivation of (11) follows analogously.

## Derivation of explicit SL and RL flow equations

In order to analyze (10) and (11), we must evaluate the following expectation values:

$$\langle \phi(\mathbf{w} \cdot \mathbf{x}) x_i \rangle_{\mathbf{x} \sim \mathcal{N}(\boldsymbol{\mu}, \boldsymbol{\Sigma})} \quad \text{and} \quad \langle \phi^2(\mathbf{w} \cdot \mathbf{x}) x_i \rangle_{\mathbf{x} \sim \mathcal{N}(\boldsymbol{\mu}, \boldsymbol{\Sigma})} \tag{32}$$

with $\phi(x) = \frac{1}{2} \left( 1 + \mathrm{Erf}\left( \frac{\sqrt{\pi}}{4} x \right) \right)$. Without loss of generality, we will calculate these integrals in a coordinate system where $\mathbf{w} = w_1 \mathbf{e}_1$. We can then factorize $p(x_1, x_2, \ldots) = p_m(x_1) p_c(x_2, \ldots | x_1)$. The marginal distribution is Gaussian with $\mu_m = \mu_1$ and $\Sigma_m = \Sigma_{11}$, and the conditional distribution is also normal with $(\mu_c)_i = \mu_i + \frac{1}{\Sigma_{11}} \Sigma_{1i}(x_1 - \mu_1)$, and $(\Sigma_c)_{ij} = \Sigma_{ij} - \frac{1}{\Sigma_{11}} \Sigma_{1i} \Sigma_{1j}$.

Furthermore, to simplify notations, we introduce $\tilde{\mathbf{w}} = \mathbf{w} \cdot \sqrt{\pi/8}$, as well as the quantities

$$a = \boldsymbol{\mu} \cdot \tilde{\mathbf{w}}, \tag{33}$$

$$b = \sqrt{\tilde{\mathbf{w}}^T \boldsymbol{\Sigma} \tilde{\mathbf{w}}}. \tag{34}$$

Let's first calculate $\langle \phi(\mathbf{w} \cdot \mathbf{x}) \rangle_{\mathbf{x} \sim \mathcal{N}(\boldsymbol{\mu}, \boldsymbol{\Sigma})}$:

$$\langle \phi(\mathbf{w} \cdot \mathbf{x}) \rangle_{\mathbf{x} \sim \mathcal{N}(\boldsymbol{\mu}, \boldsymbol{\Sigma})} = \langle \phi(w_1 x_1) \rangle_{\mathbf{x} \sim \mathcal{N}(\boldsymbol{\mu}, \boldsymbol{\Sigma})}$$

$$= \frac{1}{\sqrt{2\pi \Sigma_{11}}} \int_{\mathbb{R}} dx_1 e^{-\frac{1}{2}(x_1 - \mu_1)^2/\Sigma_{11}} \phi(w_1 x_1)$$

$$= \frac{1}{\sqrt{2\pi}} \int_{\mathbb{R}} du\, e^{-u^2/2} \phi\left( w_1(\mu_1 + u\sqrt{\Sigma_{11}}) \right) \tag{35}$$

$$= \Phi\left( \frac{a}{\sqrt{1+b^2}} \right),$$

where $\Phi(x) = \frac{1}{2} + \frac{1}{2} \mathrm{Erf}(x/\sqrt{2})$ is the cumulative distribution function of the standard normal distribution. To evaluate the last line of this and the following integrals, we used the reference [Owen, 1980].

For the integral $\langle \phi(\mathbf{w} \cdot \mathbf{x}) x_i \rangle_{\mathbf{x} \sim \mathcal{N}(\boldsymbol{\mu}, \boldsymbol{\Sigma})}$, we first do the calculation for $i = 1$:

$$
\begin{aligned}
\langle \phi(\mathbf{w} \cdot \mathbf{x}) x_1 \rangle_{\mathbf{x} \sim \mathcal{N}(\boldsymbol{\mu}, \boldsymbol{\Sigma})} &= \langle \phi(w_1 x_1) x_1 \rangle_{\mathbf{x} \sim \mathcal{N}(\boldsymbol{\mu}, \boldsymbol{\Sigma})} \\
&= \frac{1}{\sqrt{2\pi \Sigma_{11}}} \int_{\mathbb{R}} dx_1 e^{-\frac{1}{2}(x_1 - \mu_1)^2 / \Sigma_{11}} \phi(w_1 x_1) \, x_1 \\
&= \frac{1}{\sqrt{2\pi}} \int_{\mathbb{R}} du \, e^{-u^2/2} \phi\left(w_1(\mu_1 + u\sqrt{\Sigma_{11}})\right) (\mu_1 + u\sqrt{\Sigma_{11}}) \\
&= \mu_1 \Phi\left(\frac{a}{\sqrt{1+b^2}}\right) + \frac{1}{\sqrt{2\pi}} \frac{\Sigma_{11} \tilde{w}_1}{\sqrt{1+b^2}} e^{-\frac{a^2}{2(1+b^2)}}.
\end{aligned}
\tag{36}
$$

For $i \neq 1$, we get

$$
\begin{aligned}
\langle \phi(\mathbf{w} \cdot \mathbf{x}) x_i \rangle_{\mathbf{x} \sim \mathcal{N}(\boldsymbol{\mu}, \boldsymbol{\Sigma})} &= \langle \phi(w_1 x_1) x_i \rangle_{\mathbf{x} \sim \mathcal{N}(\boldsymbol{\mu}, \boldsymbol{\Sigma})} \\
&= \frac{1}{\sqrt{2\pi \Sigma_{11}}} \int_{\mathbb{R}} dx_1 e^{-\frac{1}{2}(x_1 - \mu_1)^2 / \Sigma_{11}} \phi(w_1 x_1) \left(\mu_i + \frac{\Sigma_{1i}}{\Sigma_{11}}(x_1 - \mu_1)\right) \\
&= \mu_i \Phi\left(\frac{a}{\sqrt{1+b^2}}\right) + \frac{1}{\sqrt{2\pi}} \frac{\Sigma_{i1} \tilde{w}_1}{\sqrt{1+b^2}} e^{-\frac{a^2}{2(1+b^2)}}.
\end{aligned}
\tag{37}
$$

Thus, for general $i$ and $\mathbf{w}$ we can write

$$
\langle \phi(\mathbf{w} \cdot \mathbf{x}) x_i \rangle_{\mathbf{x} \sim \mathcal{N}(\boldsymbol{\mu}, \boldsymbol{\Sigma})} = \mu_i \Phi\left(\frac{a}{\sqrt{1+b^2}}\right) + \frac{1}{\sqrt{2\pi}} \frac{(\boldsymbol{\Sigma} \tilde{\mathbf{w}})_i}{\sqrt{1+b^2}} e^{-\frac{a^2}{2(1+b^2)}}.
\tag{38}
$$

We next calculate

$$
\begin{aligned}
\langle \phi^2(\mathbf{w} \cdot \mathbf{x}) \rangle_{\mathbf{x} \sim \mathcal{N}(\boldsymbol{\mu}, \boldsymbol{\Sigma})} &= \langle \phi^2(w_1 x_1) \rangle_{\mathbf{x} \sim \mathcal{N}(\boldsymbol{\mu}, \boldsymbol{\Sigma})} \\
&= \frac{1}{\sqrt{2\pi \Sigma_{11}}} \int_{\mathbb{R}} dx_1 e^{-\frac{1}{2}(x_1 - \mu_1)^2 / \Sigma_{11}} \phi^2(w_1 x_1) \\
&= \Phi\left(\frac{a}{\sqrt{1+b^2}}\right) - 2T\left(\frac{a}{\sqrt{1+b^2}}, \frac{1}{\sqrt{1+2b^2}}\right),
\end{aligned}
\tag{39}
$$

where $T$ stands for Owen's $T$ function.

Analogously, we can calculate

$$
\begin{aligned}
\langle \phi^2(\mathbf{w} \cdot \mathbf{x}) x_i \rangle_{\mathbf{x} \sim \mathcal{N}(\boldsymbol{\mu}, \boldsymbol{\Sigma})} = \\
\mu_i \langle \phi^2(\mathbf{w} \cdot \mathbf{x}) \rangle + \frac{2(\boldsymbol{\Sigma} \tilde{\mathbf{w}})_i}{\sqrt{2\pi}\sqrt{1+b^2}} \Phi\left(\frac{a}{\sqrt{1+b^2}\sqrt{1+2b^2}}\right) e^{-\frac{a^2}{2(1+b^2)}}.
\end{aligned}
\tag{40}
$$

## Fixed point analysis

In this section, we analyze the fixed points of the systems of equations (18) & (19) for SL and (22) & (23) for RL. We will first show the intuitive result that any fixed point $\langle \mathbf{w}^* \rangle$ is maximally aligned with $\boldsymbol{\mu}$, i.e. $\langle \mathbf{w}^* \rangle \cdot \boldsymbol{\mu} = |\boldsymbol{\mu}| \cdot |\langle \mathbf{w}^* \rangle|$, as long as $\sigma > 0$. For simplicity, we set the regularization parameter $\lambda = 0$. Note that for both SL and RL, the flow equations take the form

$$
\begin{aligned}
\frac{d}{dt} \langle \mathbf{w} \rangle \cdot \boldsymbol{\mu} &= |\boldsymbol{\mu}|^2 f_1(\boldsymbol{\mu}, \mathbf{w}, \sigma) - \boldsymbol{\mu} \cdot \langle \mathbf{w} \rangle f_2(\boldsymbol{\mu}, \mathbf{w}, \sigma), \\
\frac{1}{2} \frac{d}{dt} |\langle \mathbf{w} \rangle|^2 &= \boldsymbol{\mu} \cdot \langle \mathbf{w} \rangle f_1(\boldsymbol{\mu}, \mathbf{w}, \sigma) - |\langle \mathbf{w} \rangle|^2 f_2(\boldsymbol{\mu}, \mathbf{w}, \sigma)
\end{aligned}
\tag{41}
$$

for some functions $f_1$ and $f_2 > 0$. Also, it's easy to see that $\langle \mathbf{w}^* \rangle \cdot \boldsymbol{\mu} > 0$. Thus, a fixed point $\langle \mathbf{w}^* \rangle$ satisfies

$$
\begin{aligned}
\frac{f_1}{f_2} &= \frac{\boldsymbol{\mu} \cdot \langle \mathbf{w}^* \rangle}{|\boldsymbol{\mu}|^2}, \\
\frac{f_1}{f_2} &= \frac{|\langle \mathbf{w}^* \rangle|^2}{\boldsymbol{\mu} \cdot \langle \mathbf{w}^* \rangle}.
\end{aligned}
\tag{42}
$$

Thus, setting these equal to one another, we find that $\langle \mathbf{w}^* \rangle \cdot \boldsymbol{\mu} = |\boldsymbol{\mu}| \cdot |\langle \mathbf{w}^* \rangle|$ and the two equations reduce to a single equation for $|\langle \mathbf{w}^* \rangle|$.

Assume without loss of generality that $|\boldsymbol{\mu}| = 1$. For supervised learning, (19) then implies that

$$0 = \frac{|\langle \mathbf{w}^* \rangle|}{2}\mathrm{Erfc}\left(\frac{|\langle \mathbf{w}^* \rangle|\sqrt{\pi}}{\sqrt{16 + 2\pi\sigma^2|\langle \mathbf{w}^* \rangle|^2}}\right) - \frac{1}{4}\frac{\sigma^2|\langle \mathbf{w}^* \rangle|^2}{\sqrt{1 + \sigma^2|\langle \mathbf{w}^* \rangle|^2\pi/8}}e^{-\frac{|\langle \mathbf{w}^* \rangle|^2\pi}{16 + 2\pi\sigma^2|\langle \mathbf{w}^* \rangle|^2}}, \qquad (43)$$

where $\mathrm{Erfc}(z) = 1 - \mathrm{Erf}(z)$ is the complementary error function.

We can factor out the common exponential asymptotics of both terms to get

$$0 = e^{-\frac{|\langle \mathbf{w}^* \rangle|^2\pi}{16 + 2\pi\sigma^2|\langle \mathbf{w}^* \rangle|^2}}\left(e^{+\frac{|\langle \mathbf{w}^* \rangle|^2\pi}{16 + 2\pi\sigma^2|\langle \mathbf{w}^* \rangle|^2}}\frac{1}{2}\mathrm{Erfc}\left(\frac{|\langle \mathbf{w}^* \rangle|\sqrt{\pi}}{\sqrt{16 + 2\pi\sigma^2|\langle \mathbf{w}^* \rangle|^2}}\right)\right.$$
$$\left. - \frac{1}{4}\frac{\sigma^2|\langle \mathbf{w}^* \rangle|}{\sqrt{1 + \sigma^2|\langle \mathbf{w}^* \rangle|^2\pi/8}}\right). \qquad (44)$$

Of the two terms in the parentheses, the first term starts at $\frac{1}{2}$ for $|\langle \mathbf{w}^* \rangle| = 0$ and is monotonically decreasing to the value $\frac{1}{2}e^{1/2\sigma^2}\mathrm{Erfc}(\frac{1}{\sqrt{2}\sigma}) \approx \frac{1}{\sqrt{2\pi}}(\sigma - \sigma^3 + \mathcal{O}(\sigma^5))$, while the second term starts at $0$ and increases to the larger value $\frac{\sigma}{\sqrt{2\pi}}$. Thus, there is a unique fixed point. For RL, uniqueness of the fixed point can be shown using an analogous argument.

This fixed point $\langle w^* \rangle$ is stable, because for SL, (14) implies that $\frac{d}{dt}\langle w \rangle = -\nabla_w\langle \mathcal{L} \rangle_L$, where $\mathcal{L}$ is the cross-entropy loss and $\langle \cdot \rangle_L$ is the average over the input distribution. Thus, $\langle \mathcal{L} \rangle_L$ is a Lyapunov function for the dynamical system. For RL, the same argument holds by the policy-gradient theorem, where $\langle \mathcal{L} \rangle_L$ additionally includes an average over the output noise.

Note that, for $\lambda = \sigma = 0$, although $|\langle \mathbf{w} \rangle|$ diverges, a calculation of $\frac{d}{dt}\log\frac{(\langle \mathbf{w} \rangle \cdot \boldsymbol{\mu})^2}{|\boldsymbol{\mu}|^2|\langle \mathbf{w} \rangle|^2}$ shows that the alignment between $\boldsymbol{\mu}$ and $\langle \mathbf{w} \rangle$ still converges to 1. This result is consistent with those of Ji and Telgarsky [2020].

## MNIST details

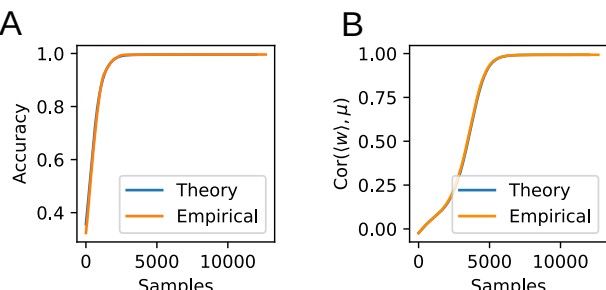

Figure 7: Comparison of the theory with training on raw MNIST. **A:** Comparison of the empirical test classification accuracy with the theoretical prediction. **B:** Just like for the Gabor-filtered inputs, the theory accurately captures non-trivial ongoing learning dynamics.

When applying our theory to the MNIST dataset, we compare SGD applied to actual data (orange curves) with calculated SGD curves from our theory (blue curves). In the main text, we preprocess the data by convolving raw pixel values with a bank of Gabor filters to approximate a more realistic scenario where the binary classifier appears at the end of a convolutional neural network. For the plots in Figure 7, we use raw pixel values to demonstrate that our evolution equations (15) work in general settings without relying on Gabor filter representation. In both cases, we globally translate all input vectors such that the dataset's mean is zero. We directly evaluate test set accuracy (Figures 5B and 7A) and the correlation of $\mathbf{w}$ at each SGD step with the mean $\boldsymbol{\mu}$ of digit '1' (Figures 5C and 7B)). To calculate the theoretical accuracy curve, we first numerically solve the differential equations (15) for the mean $\boldsymbol{\mu}$ and covariances $\Sigma_{0,1}$ obtained from the empirical dataset to derive $\mathbf{w}(t)$. We

then integrate two multivariate normal distributions with these $\boldsymbol{\mu}$ and $\boldsymbol{\Sigma}$ values in the half-spaces bounded by $\mathbf{w}(t)$ (expressible as an error function) and plot the result as the theoretical accuracy curve in Figures 5B and 7A. The tSNE embedding in Figure 5A is included for illustrative purposes only and is not used in calculations.

## Experiments

The numerical code implementing the model and performing the analyses was mostly written in JAX [Bradbury et al., 2018], as well as Wolfram Mathematica and SciPy [Virtanen et al., 2020]. For Fig. 2, the flow fields were plotted for the limit of zero input noise and a regularization parameter of $\lambda = 0.1$. The learning curves are plotted for $\lambda = 0$ and $\Sigma = \sigma^2 \mathbb{I}$, with $\sigma = 0.1$ and $\sigma = 1$. As in all other figures besides Fig. 5, we set $|\boldsymbol{\mu}| = 1$. Each experiment was repeated for 10 runs with different random seeds, with the standard deviation indicated as a (barely visible) shaded region. The curve of $|\langle \mathbf{w}^* \rangle|$ is plotted for the limit of zero input noise. For Fig. 3, we numerically integrated (10) with $\lambda = 0$. For the total variance plotted in Fig. 4, we set $\lambda = 0.1$ and integrate the differential equations (11) numerically. For Fig. 5, the Gabor-filtered inputs were created using [Haghighat et al., 2015] with default parameters. The perceptron was trained using SGD with a logistic sigmoid output, and the orange curve in panel B shows test accuracy. The training was repeated 10 times with shuffled data. We set $\lambda = 1$. For the forgetting curves in Fig. 6, we set $\lambda = 10$ and the learning rate to $\eta = 10^{-2}$. The curves shown are averages over 50 different random initializations each. The input dimension is set to $N = 500$. For all other simulations, the learning rate was set to $\eta = 10^{-3}$. The computations were performed on an NVIDIA Titan Xp GPU, with runtimes of at most a few minutes.

