# OpenReview forum: "Dynamics of Supervised and Reinforcement Learning in the Non-Linear Perceptron"
_NeurIPS.cc/2024/Conference — NeurIPS 2024 poster_

### Official Review · Reviewer_m7rp · 2024-07-08

**Soundness:** 3
**Presentation:** 2
**Contribution:** 2
**Rating:** 7
**Confidence:** 4

**Summary:**

In this work, the authors derive a set of equations describing gradient flow in a non-linear finite-dimensional perceptron under the assumptions that the data is multinormally distributed, there is a small learning rate, and the task is binary classification. They develop the equations both for the case of a supervised learning rule and for a reinforcement learning rule. From these equations and simple simulations, they study the impact of noise on learning time, the influence of anisotropy in input distributions, input noise covariance and continual learning. They furthermore use a preprocessed MNIST dataset to asses their derived equations beyond toy datasets.

**Strengths:**

This paper adresses an important open question, namely how we should study or understand the learning dynamics of non-linear neural networks. The mathematical derivations seem sound and correct, the paper is well-written, and the carefully designed figures not only illustrate the results but also aid in understanding the approach (cfr figure 1). The theory has been developed for both the supervised and reinforcement learning case. The authors have made an excellent effort to demonstrate the insights that can be gained from their mathematical derivations, and have extended their experiments beyond toy datasets to illustrate the vailidity of their approach further.

**Weaknesses:**

The work is seemingly presented as a general theory of learning in non-linear perceptrons; however, the assumptions are very strong and far from practical settings. The assumptions might even render the learning dynamics equivalent to those of a linear model (see questions). This should be more clear in abstract and/or introduction.

It is hard to assess the novelty of the results, as the work seems to lack references to closely related works that derive equations in similar settings and/or draw similar conclusions. Notably a comparison to the work by Refinetti [5] would be beneficial (see questions).

Far more relevant related work exists on the dynamics of learning in complex and/or non-linear networks. Currently, the introduction presents the related work as merely falling into two approaches: the student-teacher setup and the linearized perceptron. However, several works exists for far more complex and even non-linear setups exist. References below are just a few of the possible additions.

The MNIST dataset is heavily preprocessed and it is mentioned that the two classes used are modfied in the following way: (L177) ‘We then model these two input classes as multivariate Gaussians with covariances Σ0, and means μ0,1 178 (or °æμ after a translation).’ It is unclear how this dataset is still a deviation from the multinormal assumption.

**Questions:**

1)	Are the insights following the derivations of the gradient flow equations (i.e., sections impact noise on training time, Anisotropic input distributions, etc), also based on the assumption that the covariance of the weights is zero (L107)? If so, please elaborate on to what degree this assumption limits the applicability of your results in practical settings.

2)	How does your work differ from the theory developed in [5]?

3)	Provided I understand [5] and your work correctly, it seems that the dynamics of learning in non-linear perceptrons when the input data is Guassian is equivalent to the dynamics of a linear model. As such, your theory would not really capture the effects of the non-linearity. Please elaborate; if this is correct, please indicate how you would rephrase the paper to reflect this fact.

4)	Please compare your insights to the analyis of Saxe [4]. Specifically, they mention an analysis of the following: we find that perceptual correlations can either speed up or slow down learning, depending on whether they are aligned or misaligned with the input dimensions governing task-relevant semantic distinctions.

5)	Please compare your insights on continual learning with the work of [3]

6)	Please explain better how you used/modeled the MNIST dataset, cfr L177.

7)	How would you expand the related works to include more relevant papers?

In general: if you agree something would be a useful contribution to your work, please explicitly state how you would update the current manuscript.

1)	Ji, Z., & Telgarsky, M. (2020). Directional convergence and alignment in deep learning. Advances in Neural Information Processing Systems, 33, 17176–17186.
2)	Saxe, A. M., Sodhani, S., & Lewallen, S. (n.d.). The Neural Race Reduction: Dynamics of Abstraction in Gated Networks.
3)	J Dominé, C. C., Braun, L., Fitzgerald, J. E., & Saxe, A. M. (2023). Exact learning dynamics of deep linear networks with prior knowledge *. Journal of Statistical Mechanics: Theory and Experiment, 2023(11), 114004. https://doi.org/10.1088/1742-5468/ad01b8
4)	Saxe, A. M., McClelland, J. L., & Ganguli, S. (2019). A mathematical theory of semantic development in deep neural networks. Proceedings of the National Academy of Sciences, 116(23), 11537–11546. https://doi.org/10.1073/pnas.1820226116
5)	Refinetti, M., Ingrosso, A., & Goldt, S. (n.d.). Neural networks trained with SGD learn distributions of increasing complexity.
6)	Pinson, H., Lenaerts, J., & Ginis, V. (2023). Linear CNNs Discover the Statistical Structure of the Dataset Using Only the Most Dominant Frequencies. International Conference on Machine Learning, ICML 2023, 23-29 July 2023, Honolulu, Hawaii, USA, 202, 27876–27906. https://proceedings.mlr.press/v202/pinson23a.html

**Limitations:**

See questions above; no ethical considerations.

---

> ### Author Response · Authors · 2024-08-07
>
> Thanks to the reviewer for their overall positive assessment of our work, including its importance, soundness, and exposition. Thanks also for the numerous constructive comments, which we have addressed as described below.
>
> The main weaknesses identified by the reviewer involve assumptions that we failed to adequately explain–particularly the Gaussianity of the input distribution and preprocessing of the MNIST dataset–as well as connections of our work to recent related literature. As we describe below, these shortcomings have been fully addressed in our updated manuscript.
>
> For a detailed discussion of the preprocessing of the MNIST dataset,please refer to the explanation in the main rebuttal.
>
> The other major issue concerned whether our theory is somehow equivalent to a linear model given that we model the input data as Gaussian. It is not, as we describe in detail in the rebuttal below.
>
> Finally, the reviewer pointed out connections of our work to other recently published work. We are grateful to the reviewer for these and discuss how these works bear on ours in detail below.

---

> ### Author Rebuttal · Authors · 2024-08-07
>
> # Questions
>
> - Setting the weight covariance to zero.
>
> We found that Cov(w) approaches 0, which is equivalent to saying that all initial conditions $w^0$ converge to the same fixed point. Because the results mentioned above do not depend qualitatively on the initial condition, including Cov(w) > 0 would effectively average over an ensemble of trajectories w(t) with different initial conditions, which would give the same qualitative behaviors that we showed since all (or almost all) of these trajectories exhibit qualitatively similar behavior.
>
> - Relationship to Refinetti et al (2023).
>
> Thanks for pointing us to this very interesting paper. In that work, the authors perform a Taylor expansion for a nonlinear perceptron performing binary classification and show theoretically that, iff the nonlinear terms in the expansion are included, the value of the weights once learning has converged depends on beyond-second-order cumulants of the input distribution.
>
> Hence, the linear perceptron is unable to fit higher-order (i.e. non-Gaussian) statistical structure in the input data.
> Our finding that the fixed point of the system depends on the nonlinearity of the activation function is perfectly consistent with these results. The reviewer’s concern that, because we only include the first two cumulants of the input-data distribution in our model, the nonlinearity of the activation function is not captured by our model, does not follow from their results. It is true, however, that our choice to include only the first two cumulants of the input-data distribution in our model is a significant limitation in applying the results of our theory. Indeed, this is the first limitation that we mention in our Discussion section. However, our results on the MNIST data show that, at least for this dataset, our theory does well at capturing the dynamics of learning for non-Gaussian input-data distributions.
>
> Apart from the issue of Gaussianity and how it relates to non-linearity of the activation function, our approach is overall quite distinct from that of Refinetti et al in the following ways: (i) they analyzed only SGD, whereas we compare RL vs. SGD; (ii) their theoretical results were obtained perturbatively, whereas our calculations are non-perturbative and fully capture nonlinear effects at all orders; and (iii) their theory focused on the properties of the fixed point that learning converges to, whereas our work additionally focuses on the dynamics of learning and how these are affected by the properties of the input distributions. In particular, the result on how input noise along or orthogonal to the coding direction impacts learning speed crucially depends on the saturating nonlinearity of the activation function, which is not captured by a perturbative calculation.
>
> In response to the reviewer’s suggestion, we have included the following statement in our Discussion: “Indeed, recent work on the nonlinear perceptron has shown that, the first- and second-order cumulants of the input distribution are learned early in training, later stages of training involve learning beyond-second-order (i.e. non-Gaussian) statistical structure in the input data (Refinetti et al, 2023), suggesting that our theory's ability to describe late-stage training in complex datasets may be somewhat limited.”
>
> - Relationship to Saxe et al (2019).
>
>  Thanks for pointing out the connection between this work and ours. In that paper, “perceptual correlations” refers to the anisotropy of the input distribution. In their Supplemental Material, Saxe et al found that, in a two-layer linear network trained with student-teacher learning to perform a linear mapping $y = \mathbf{w}^* \cdot \mathbf{x}$, training converged more quickly when the input data had large variance along $\mathbf{w}^*$. This is interesting to compare it with our results, since we found that increased input variance along the coding direction *decreases* the speed of learning.
>
>
> In the updated version, we have included the following remark below Eq. (25) to account for the difference:
> “This is in apparent contrast to student-teacher learning in two-layer networks, where input variance along the task-relevant dimension tends to *increase* the speed of learning (Saxe et al, 2019). The reason for these seemingly opposite results is because, in the student-teacher case, variance along the coding direction is a signal that facilitates learning, while, in our case of binary classification, variance along the coding direction is noise that impairs learning.”
>
> - Relationship to Domine et al (2023).
>
> The approach of Domine et al (2023) differs from ours in that their work analyzes two-layer networks in the student-teacher setup, whereas ours analyzes a single-layer nonlinear network performing binary classification. While continual learning was not the main focus of that work, they analyze a continual-learning setup within their framework, with the main finding being that nonlinearity in the network increases the amount of catastrophic forgetting. Comparing linear vs. nonlinear versions of our model to see whether this result also holds in a single-layer classifier could be interesting, but this is not a direction that we pursued.
>
> In response to the reviewer’s suggestion, we have added the following to our Discussion: ``Recent work has found that nonlinearity can drastically increase the amount of catastrophic forgetting in continual learning (Domine et al, 2023), so the nonlinearity in our classifier may play a significant role here as well.”
>
> - Relationship to Ji et al (2020).
>
> We thank the reviewer for pointing us to this great reference. In this work, the authors prove for a large class of (generally non-linear) models that weights converge in direction, even in situations where their size diverges. Motivated by this work, we showed that for our model, this result explicitly follows from the flow field equations and cited the paper.
>
> - MNIST approach: See main rebuttal.

---

> > ### Comment · Reviewer_m7rp · 2024-08-12
> >
> > I have read the rebuttal, The experiments with the raw MNIST data are convincing, and the added discussion and comparisons with related works strengthen the paper considerably. I would like to thank the authors for their effort and clarifications.
> >
> > I have one last remark: the work of (Saxe et al, 2019) and (Domine et al, 2023) is not based on the student-teacher setup. Cfr. in (Domine, 2023):  "A line of theoretical research has considered online learning dynamics in teacher-student settings [ 36 , 37, 38 ], deriving ordinary differential equations for the average learning dynamics even in nonlinear networks. However, solving these equations requires numerical integration. By contrast, our approach provides explicit analytical solutions for the more restricted case of deep linear networks." Please consider this for the camera-ready version.
> >
> > I am by now convinced this is a solid contribution to the NeurIPS conference with a moderate to high impact on the field of theoretical deep learning, and I will raise my score to a 7.

---

> > > ### Author Response · Authors · 2024-08-13
> > >
> > > We thank the reviewer for providing us with all the missing citations, and we've fixed the incorrect references to these two papers. We are very grateful for the careful consideration of our submission.

---

### Official Review · Reviewer_SUcs · 2024-07-13

**Soundness:** 3
**Presentation:** 2
**Contribution:** 4
**Rating:** 7
**Confidence:** 3

**Summary:**

This paper provides what seems to be the first derivation of equations for the dynamics of SGD-style learning in a nonlinear perceptron outside of the standard student-teacher paradigm. Specifically, a Fokker-Planck-like PDE is derived for the temporal dynamics of the weight distribution and from this ODEs are derived for the first and second moments of the weights. The equations can be applied generally to different learning rules; in the paper supervised learning of logistic regression and REINFORCE are studied. Through numerical solving and analysis of the dynamical equations several interesting results are found. In particular, it was observed that increasing noise has opposite effects on supervised learning compared with REINFORCE, and that noise orthogonal to the classification boundary speeds learning in both REINFORCE and supervised learning.

**Strengths:**

In my view this paper’s strengths are its impact and novelty. As far as I can tell there are no other works that derive the learning dynamics for nonlinear perceptrons except for in a student-teacher regime, making this work novel. Because nonlinear perceptrons comprise the building blocks of many deep learning algorithms, a derivation of learning dynamics for the nonlinear perceptron is, in my view, rather valuable (to the point that it is surprising that this has not been done already!). The paper also goes beyond just deriving equations, but studies them and finds some interesting results related to how noise impacts learning.

**Weaknesses:**

In my view the paper’s primary weaknesses is its clarity. There are many points in the paper where equations are derived and the sub-steps are not thoroughly explained. This might be less of a problem in a physics or mathematics publication, but for the general audience encountered at NeurIPS I believe that there should be more “holding of the reader’s hand” w.r.t. derivations. There are also several sections where it could be useful to explain assumptions a little more thoroughly (e.g. around the continuum limit). If it wasn't for this point regarding assumptions I would have given the paper a soundness rating of 4. If it wasn't for the issues of clarity I would have given a higher presentation rating.

# Notes on my review
In part because of this issue of clarity–and in part because of lack of time on my part and a large reviewing load for this conference–I have been unable to verify many of the equations in the paper. Specifically, I have only verified the math up to and including Equation 9. Thus, it is difficult for me to comment on the soundness of many of the results, leading to my low confidence score. Critically, my score for the paper assumes that the math that I have not verified is sound.

**Questions:**

I am arranging this section in bullet points for ease of reading. Properly addressing these questions/comments could lead to me improving my rating of the paper during the rebuttal. I have divided these questions/comments based on whether I perceive them to be highly important (Primary), or regularly important (Secondary). I have also included a section with basic writing related comments at the end (Syntax). When a question/comment corresponds with a line (or lines) of the paper I list that line at the start of the comment.

# Primary
- Previous work studying SGD dynamics has had trouble deriving a continuum limit that meaningfully takes into account noise (i.e. that doesn’t become gradient dynamics in the limit – see e.g. section 2.3.3 in Yaida 2019, a reference in your paper). It could be nice to discuss a little (beyond the comment on line 70) in the text how your work is able to derive a continuous time F-P equation and avoid this issue
- Related to the above: it would be nice to discuss the assumptions necessary for the continuum limit in the main derivation. If there are key assumptions, this could be listed in the limitations in the discussion section
- 78 (equation 9): this is of course basically a Fokker-Planck equation but with a particularly chosen drift term and no diffusion term. Could you elaborate on the similarities and differences with the Fokker-Planck Equation (FPE) please?
- 80: I am not sure why the definition $\delta w = \langle w \rangle - w$ can be made. Is this a different variable than $\delta\mathbf{w}$ (with $w$ boldface) from before?
- 80-81: Risken is referenced as the means of deriving the moment equations. Given that the PDE studied is a very particular version of the FPE, and that NeurIPS represents a fairly general audience, it would be nice to include derivations of the moment equations. These could be included in the appendix. It would also be nice to have page numbers for where the referenced derivations in Risken appear
- 125-126: it is claimed that there is a unique globally stable fixed point, but this is not immediately clear to me from the equations. Perhaps the analysis of the equations leading to this conclusion could be listed in the appendix?

# Secondary
- It could be nice to discuss how your work compares with the related work on SGD dynamics: *Homogenization of SGD in high-dimensions: Exact dynamics and generalization properties* – Paquette et al. 2022.
- 59: might be nice to remind the reader here that $w \in \mathbb{R}^n$ for any $n$ (unless I am mistaken). This is an important aspect of the paper and deserves to be highlighted.
- 67: would suggest changing “Performing a Taylor expansion” to a sentence that describes that you are changing the variable of integration from $w’$ to $\delta w$ and then Taylor expanding w.r.t. $\delta w$. This doesn’t take much extra space and is helpful for the reader.
- 67 (equation 4): it would be good to include the smoothness assumptions on $p$ that are required to interchange the order of integration and differentiation in this equation
- 131: “particularly for SL” => “only for SL”.
- 175: is the convolution of the input data with Gabor filters necessary? If so, a little elaboration on why would be useful
- Figure 6: not immediately clear which of the two plots are derived from the flow equations and which are from numerical simulations. Could this be explained?
- 223-224: it is suggested that this approach would allow for effects due to finite learning rate, but the learning rate is taken to zero for the continuum limit, no? Does this future direction involve simply working in discrete time?
- 321: why was $\lambda$ set to $10$ for the forgetting task? This seems abnormally high

# Syntax
- Fig 1.B: might suggest replacing multinormal => multivariate normal as the latter seems more conventional
- 34-39: long sentence. Suggest ending at “... (e.g., supervised and reinforcement 36 learning)” <period>. And starting a second sentence. For the second sentence, avoid saying both “important goal” and “longstanding goal”, for more concise writing
- 67: “becomes” => “yields” or similar
- 110 (equation 15 line 1): parentheses is on right side of $x_i$ when it should be on left
- 77: would suggest defining the average w.r.t. $p(w,t)$ later (e.g. line 83) as it doesn’t show up in the equations yet.

**Limitations:**

In this section I will primarily discuss issues related to the NeurIPS checklist. I will do so in point form in the order of appearance of the topics in the checklist. Below, I only include bullet points where I have concerns about what the authors list. Addressing these issues could also lead me to improve my rating of the paper.

- Theory assumptions and proofs:
  - I disagree with the justification for this section. Certain assumptions are not included (e.g. assumptions necessary to interchange differentiation and integration mentioned above) and extra insight could be provided on derivations (see “clarity” mentioned above).
- Broader impact:
  - I would argue that, even with theoretical work, one should consider downstream interests. E.g., is it ethical to study general algorithms that can easily be employed for harm (e.g. in weapons industry) without taking steps to support policy that restrict the negative application of such innovations?
  - Safegaurds: this should be “No” instead of “N/A”, for the above reasons.

---

> ### Author Response · Authors · 2024-08-07
>
> We appreciate the reviewer’s assessment of our work as impactful and novel. We also appreciate the reviewer’s constructive suggestions to improve our paper’s clarity, which was described as the primary weakness of the paper. In particular, the reviewer pointed out that our work’s soundness and presentation would be enhanced by filling in more steps of our derivations and by explaining certain assumptions more thoroughly. In response to the reviewer’s suggestion, we have written a new appendix in which we provide the details of our derivations of the moment equation (10) and (11), the flow equations, as well as another appendix in which we prove the existence of a unique, stable fixed point. We also explicitly added mathematical assumptions where they are being used. We describe additional changes we have made to address the reviewer’s comments in the rebuttal.

---

> ### Author Rebuttal · Authors · 2024-08-07
>
> # Primary questions and comments
>
> - Flow equation derivation. In deriving our continuous-time flow equation, we don’t make a priori assumptions about the nature of the noise, but rather derive the FPE-like equation directly from the weight-update equation. Although this is not yet realized in the current work, a motivation for this approach is the non-Gaussian noise inherent in non-linear models even for Gaussian inputs, as well as in features of deep neural networks. While, at leading order in $\eta$, the evolution equation for the mean of the weights that we arrive at is equivalent to gradient flow, the method by which we arrive at this equation is different from most other works in the literature. In addition, our approach yields a flow equation for the weight covariance, which is not present in gradient-flow approaches.
>
> - Assumptions for the continuum limit. While the equations for finite learning rate assume that the discrete process $\mathbf{w}$ can be interpolated smoothly between updates, with constraints on the uniform convergence properties of the series expansion, these assumptions are easily satisfied in the $\eta\to 0$ limit as long as the process $f_i(w)$ in the weight update equation behaves nicely in the limit. We assume that the moments of $f$ are finite and smooth, and we have explicitly added this assumption in the paper.
>
> - Relationship to the Fokker-Planck equation. At the order to which we are working, the reviewer is correct that our Eq. (9) is an FPE with no diffusion term. Under certain mathematical assumptions, one can work at the next order in $\eta$. Then the right hand side of the equation would be identical to that of an FPE, while the left hand side would contain a second-order time derivative. This approach provides an alternative to previous approaches to analyzing the effects of nonzero learning rate and is the subject of our forthcoming paper.
>
> - Definition of $\delta w$. Thanks to the reviewer for pointing out the potential point of confusion about the variable $\delta w$. In the revised version, we have not defined this variable in the main text, and, where we use it in the appendix, we define it instead as $\hat{w}$ in order to avoid confusion with the variable $\delta \mathbf{w}$.
>
> - Existence of a unique, stable fixed point. In response to the reviewer’s suggestion, we have added a new appendix in which we explicitly prove the existence of a unique, stable fixed point in our flow equations.
>
> # Secondary questions and comments
>
> - In Paquette et al., the authors study regularized least-squares regression and prove that they can approximate the loss function for finite learning rates by the loss function of the solution of a stochastic differential equation in the limit of high input dimensions. We thank the reviewer for pointing out this reference. In the present work, we study a somewhat different setup and don’t include finite-$\eta$ effects, but the mentioned result can serve as a valuable point of comparison for the theory with finite $\eta$ in an upcoming publication.
>
> - Interchanging the order of differentiation and integration. We thank the reviewer for pointing this out. We added technical requirements on $p(\mathbf{w}+\delta \mathbf{w}, t+\delta t|\mathbf{w},t)$ to apply the dominated convergence theorem. For the update equations and noise distributions considered in this paper, this requirement is satisfied.
>
> - Preprocessing of the MNIST data with Gabor filters. Please see the explanation and figure in the main rebuttal.
>
> - Effects of nonzero learning rate. Please see our response to the question above about the relationship of our approach to the Fokker-Planck equation.
>
> - Value of $\lambda$ in the forgetting task.
> The (logarithmic) range of input noise levels between the level where noise is negligible to the point where forgetting happens too fast to show on the plot is rather small. Since higher values of lambda lead to faster convergence, and we wanted to include a high number of runs to average over, we decided to set $\lambda=10$ for this plot to increase the speed of our simulations. To address the reviewer's question, we have repeated this procedure for $\lambda=1$ and can produce a figure very close to Figure 6B for the noise values $\sigma=0.2$ and $10^{-2}$.
>
> All of the other suggestions made in the reviewer’s list of Secondary Questions and Comments have been incorporated into the revised version of our paper.
>
> # Syntax suggestions
>
> Thanks to the reviewer for these helpful suggestions. They have all been implemented in the updated version of our manuscript.
>
> # Limitations
>
> - Statements of assumptions and details of derivations. We believe that our revisions, including the new appendices, described above have addressed all of the reviewer’s comments on these points.
>
> - Broader impact and safeguards. While we appreciate the reviewer’s conscientiousness about the broader impacts of basic research and agree that such issues are important, the Checklist guidelines seem to state clearly that including a statement about broader impacts and safeguards in papers focused on basic research questions is not appropriate for the type of work that we have submitted. The example given is the following: “It is not needed to point out that a generic algorithm for optimizing neural networks could enable people to train models that generate Deepfakes faster”. While we recognize that reasonable people could differ on this point, our view is that adding a boilerplate statement on AI safety to every single paper, regardless of how relevant or irrelevant the point may be to the work, would detract from attention being paid to such statements in papers where broader impacts and safeguards are actually directly relevant.

---

> > ### Comment · Reviewer_SUcs · 2024-08-12
> >
> > Thank you to the authors for their detailed responses; I have increased my score accordingly. While I still have some uncertainty regarding the paper on account of my own failing to check every detail of the math, from the aspects that I have understood this paper seems impactful and entirely worthy of publication at NeurIPS.

---

> > > ### Author Response · Authors · 2024-08-13
> > >
> > > We thank the reviewer for the adjustment, and we hope that the updated version of our paper will help elucidate the missing details of the mathematical derivations.

---

### Official Review · Reviewer_WrR8 · 2024-07-19

**Soundness:** 3
**Presentation:** 2
**Contribution:** 2
**Rating:** 5
**Confidence:** 2

**Summary:**

This paper analyzes the weight dynamics of single layer neural networks with nonlinear output functions.

**Strengths:**

The model of stochastic weight evolution dynamics is quite general.

**Weaknesses:**

The theory is tested only in binary classification task.

**Questions:**

Why is the probabilistic output is considered as reinforcement learning? It should be driven by reward signal.

**Limitations:**

It is not clear hoe this extends to multi-layer cases.

---

> ### Author Rebuttal · Authors · 2024-08-07
>
> We thank the reviewer for their feedback, but we disagree with their conclusion that our paper should not be accepted based on the fact that our theory is applied only to binary classification in a single-layer system. Indeed, according to the other reviewers, our work “addresses an important open question” (m7rp), and “This paper’s strengths are its impact and novelty… Because nonlinear perceptrons comprise the building blocks of many deep learning algorithms, a derivation of learning dynamics for the nonlinear perceptron is, in my view, rather valuable” (SUcs).
>
> We recognize that there is a tradeoff in developing theories of simple systems, where relatively thorough understanding of mechanisms can be obtained, vs. complex systems, where obtaining such understanding is difficult without making drastic approximations. In the past, NeurIPS has recognized the value of both approaches. As an example of work in a relatively simple setting similar to the one that we employ, we cite the work of Mignacco et al (NeurIPS, 2020), who derived a theory of SGD dynamics in a single-layer network with a linear or quadratic activation function performing binary classification.
>
> If the reviewer has any specific critiques of our paper’s soundness, presentation, or contribution, we would be happy to discuss and address them.
>
> Finally, regarding the question that the reviewer asked, the learning is indeed driven by the reward signal in our reinforcement-learning setup, where the reward is $\pm 1$ depending on whether the binary output is correct or incorrect (see below Eq. (13)). As usual in RL, the stochastic output may lead to greater or less reward, and the parameters are updated to maximize this reward.

---

### Author Rebuttal · Authors · 2024-08-07

We are grateful to the reviewers for their comments and suggestions on our work. We were encouraged by their recognition of the work’s novelty and impact, as well as its soundness and exposition. We have made numerous revisions to our paper in order to address the reviewers’ critiques, the three most significant of which we summarize here.

The first main substantive issue that came up in the reviewers’ critiques was the need to provide additional details on intermediate steps in our mathematical derivations. In response, we have added new appendices to the paper that fill in the details of most of our calculations. In the first section of the appendix, we derive the moment equations (10) and (11). In the second part, we derive the flow equations (15) and (20) by explicitly calculating the involved integrals. Furthermore, we analyze the convergence of these equations and provide a detailed proof of the existence of a unique, stable fixed point. We thank the reviewers for encouraging us to provide these additional details, and we expect that the addition of these appendices will make it much more straightforward for future readers to follow and reproduce our results.

The second main substantive issue brought up by the reviewers was the need to be more explicit about certain assumptions in our empirical results and mathematical derivations. In applying our theory to the MNIST dataset, we did not properly explain our motivation for preprocessing the input data with a bank of convolutional Gabor filters, leading to the reasonable suspicion that our theory, which models the input-data distributions as Gaussian, could not be successfully applied to realistic datasets. Our motivation for initially passing the MNIST data through a layer of untrained, convolutional Gabor filters was that, in practice, a binary classifier such as the one that we model would often appear at the output end of a neural network rather than being applied to raw input data, so that preprocessing the data in this way brings our model closer to a realistic use case.

Thus, our motivation in performing this preprocessing was not to make the input data more Gaussian, though it may have had that effect. In response to the reviewers questions, we have rerun this experiment using the raw MNIST input data, without the convolutional preprocessing. The results are nearly identical to the ones we showed and are included in the appendix, as well as attached to this rebuttal. This provides further evidence that our approach in practice successfully characterizes the dynamics of learning in cases where the input data is non-Gaussian.

To be as transparent as possible, here is a detailed explanation of how the MNIST Figure 5 was produced:

“When applying our theory to the MNIST dataset, we make the comparison between SGD applied to the actual data without any modeling (the orange curves) and the calculated SGD curves obtained from our theory (the blue curves). For the empirical curves, we perform SGD on either the raw pixel values for the plots in the appendix, or on a representation obtained by convolving the raw pixel values with a bank of Gabor filters in the main text, with the only modification being a global translation of all input vectors to zero the global mean of the dataset. We then simply evaluate the accuracy on the test set for Figure 5B, and the correlation of $\mathbf{w}$ at each SGD step with the mean $\mathbf{\mu}$ of the digit ‘1’ for Figure 5C. To calculate the theoretical accuracy curve, we first numerically solve the differential equations (15) for the mean $\mathbf{\mu}$ and covariances $\Sigma_{0,1}$ calculated from the empirical dataset to obtain $\mathbf{w}(t)$. We then integrate two multivariate normal distributions with these values of $\mu$ and $\Sigma$ in the half-spaces bounded by $\mathbf{w}(t)$ (which can be explicitly expressed as an error function) and plot the result as the theoretical accuracy curve in Figure 5B and find a perfect agreement with the empirical SGD curve. The tSNE embedding in Figure 5A is not used for any calculations, but only drawn for illustrative purposes.”

In addition to these changes, as described in detail below, we have added numerous minor clarifications to the text to clarify the assumptions and reasoning in our mathematical derivations, with the main restriction being the existence of all higher moments of the weight update term $\langle f_i^k\rangle_L$, $k=1,2,\ldots$ as smooth functions of $\mathbf{w}$. All these assumptions are satisfied for the non-linear perceptron we apply our results to.

Finally, the reviewers pointed out connections of our work to other recently published work on learning dynamics. Of particular note was a concern that, because we model the input-data distributions as Gaussian, the learning in our nonlinear model might be equivalent to that in a linear model. In our response below, we have provided a detailed explanation of why this is not the case. Overall, we do not believe that the existence of any of these other works detracts significantly from our work’s novelty, and we are grateful to the reviewers for giving us the opportunity to draw connections with relevant work from the literature that had escaped our attention.

In the responses that follow, we provide detailed responses to the reviewers’ questions and comments. We believe that their suggestions have been completely addressed in our revised version.

---

### Decision · Program_Chairs · 2024-09-25

**Decision:**

Accept (poster)

**Comment:**

The paper provides a theoretical study of the behavior of a nonlinear perception on a simple problem.

Reviewers recommended Accept, Accept and Borderline reject. While the settings are restricted, the contribution seems valuable. Reviewers appreciated the authors' modifications in response to reviews.

Overall I am willing to accept the paper.